# TOWARDS POLYHEDRAL AUTOMATIC DIFFERENTIATION

A PREPRINT

**Jan Hückelheim**
Imperial College London
London, UK
j.hueckelheim@imperial.ac.uk

**Navjot Kukreja**
Imperial College London
London, UK
n.kukreja@imperial.ac.uk

September 17, 2019

## ABSTRACT

Most Automatic Differentiation (AD) tools lack a way to explicitly represent or differentiate performance-critical and hardware-dependent constructs such as parallelism, vectorisation, or tiling. Machine-learning frameworks work around this by hiding implementation details from the AD process, but lack the generality of general-purpose programming languages. Instead, this talk discusses the polyhedral model as a way for general-purpose AD tools to preserve performance tweaks through the differentiation process.

***Keywords*** Automatic Differentiation · Polyhedral Compiler · Parallelisation · Vectorisation

## 1 Introduction

Derivatives are an important ingredient for optimisation, inverse modelling, error analysis, and training in scientific computing and machine learning. AD tools consume the implementation of a differentiable function, referred to as *primal*, and produce a new program that computes the derivative of the primal. AD has been implemented in the past using operator overloading [1, 2], source-to-source [3, 4, 5, 6], or just-in-time compilation [7, 8].

Machine learning applications rely on parallelism, vectorisation, and optimisations such as tiling to obtain high performance on CPUs, GPUs, or TPUs. This is increasingly also true for scientific applications. Using automatic differentiation (AD) or back-propagation for training is only practical if the derivatives are also computed efficiently on these processors.

Machine learning frameworks encapsulate performance-critical implementation details inside operators that work on large objects (e.g. Tensors). Users and AD tools alike can obtain high performance by writing programs in terms of these highly efficient operators. The framework then assembles them into larger programs. Because each operation handled by the AD tool is itself relatively costly, any additional overhead introduced by the AD process is amortised.

This model works very well when the number of operators allowed is small, and the code is optimised for a small set of target hardware. As the number along both these dimensions (number of operators allowed, number of hardware platforms targeted) increases, the development and maintenance of the underlying framework code becomes a major effort. Hence, there is a need for the user of these frameworks to be able to create their own operators at a finer-grained level of operations, while still being able to automatically differentiate and obtain good performance.

In contrast, AD tools for general-purpose programming languages are not restricted to a particular framework and naturally offer full flexibility, but have to work on a lower level of abstraction where the computation is not cleanly separated from hardware-specific implementation details. As a result, derivatives often run inefficiently or in some cases even incorrectly on modern hardware.

Section 2 briefly outlines some challenges that AD tools need to overcome when dealing with performance-optimised programs. Following this, section 3 discusses the role of the intermediate representation in an AD tool when differentiating such programs. Section 4 then shows PerforAD, a tool that we have previously presented and that uses the polyhedral model to overcome some of these challenges, before we conclude in section 5.

## 2 Background

Preserving performance-optimisations during differentiation is challenging for many reasons, two of which are given in this section.

The differentiation process creates auxiliary variables to store derivative values. The memory footprint of the derivative program is therefore usually larger than that of the original program. Implementation choices such as cache block sizes in the primal code are often not ideal for the derivative code.

Reverse-mode automatic differentiation or back-propagation works by accumulating derivatives backwards through a program. This so-called *data-flow reversal* means that every time the original program performs a read access, the derivative program will instead perform a write access at a corresponding location. This means that some thread-safe primal programs that contain only concurrent read access to shared memory locations will not be thread-safe after differentiation, as they contain concurrent write accesses [9, 10].

## 3 Internal Representation

AD tools commonly work with one of two internal representations of the original program – *operator-overloading-*style tools create a tape at run-time, which contains an unrolled sequence of scalar numerical operations, while *source-transformation*-style tools create at compile-time a control flow graph, which contains basic blocks of program instructions that are individually differentiated. Both of these representations do not capture well shared-memory parallelism, or SIMD/SIMT vectorisation that may have been present in the program. Moreover, the fact that basic blocks are differentiated individually can preclude parallelisation of the derivative code. We will show the reasons for these disadvantages in more detail during our talk.

A tool that can preserve parallelism, vectorisation and other performance optimisations will require a holistic view of loop nests. The polyhedral model is an elegant way to represent loop nests and branches contained therein, by modelling the loop body as a set of statements that are each applied to a subset of the iteration space, and by modelling the control flow as a polyhedron. This works for loops whose bounds are given as affine functions.

## 4 Case study: PerforAD

In previous work we presented PerforAD [11], an AD tool that transforms input programs performing easily parallelisable stencil operations, into output programs that compute the derivative of that stencil operation using the same stencil-like data flow pattern. PerforAD uses a polyhedral model to capture and modify iteration spaces, and performs a pre-programmed sequence of loop transformations to produce a parallelisable derivative program. This section explains the transformations step by step using a one-dimensional example. This explanation is accompanied by the illustration in Figure 1. In our previous paper, we demonstrated a full procedure for arbitrarily deep loop nests, as well as an open-source implementation that executes the transformations.

Suppose that a primal program contains the following parallel gather operation for an iteration space $i \in [1, n-1]$:

```
#pragma omp parallel for private(i)
for ( i=1; i<=n - 1; i++ ) {
  r[i] = c[i]*(2.0*u[i-1]-3.0*u[i]+4*u[i+1]);
}
```

A straightforward reverse-mode differentiation of this loop to compute the derivatives of r with respect to u would yield the following scatter operation, where ub and rb represent the adjoint variables that correspond to u and r:

```
for ( i=1; i<=n-1; i++ ) {
  ub[i-1] += 2.0 * c[i] * rb[i];
  ub[i]   -= 3.0 * c[i] * rb[i];
  ub[i+1] += 4.0 * c[i] * rb[i];
}
```

If we assume that floating-point summation is associative, we can split this into three loops.

```
for ( i=1; i<=n-1; i++ ) {
  ub[i-1] += 2.0 * c[i] * rb[i];
}
for ( i=1; i<=n-1; i++ ) {
  ub[i]   -= 3.0 * c[i] * rb[i];
}
for ( i=1; i<=n-1; i++ ) {
```

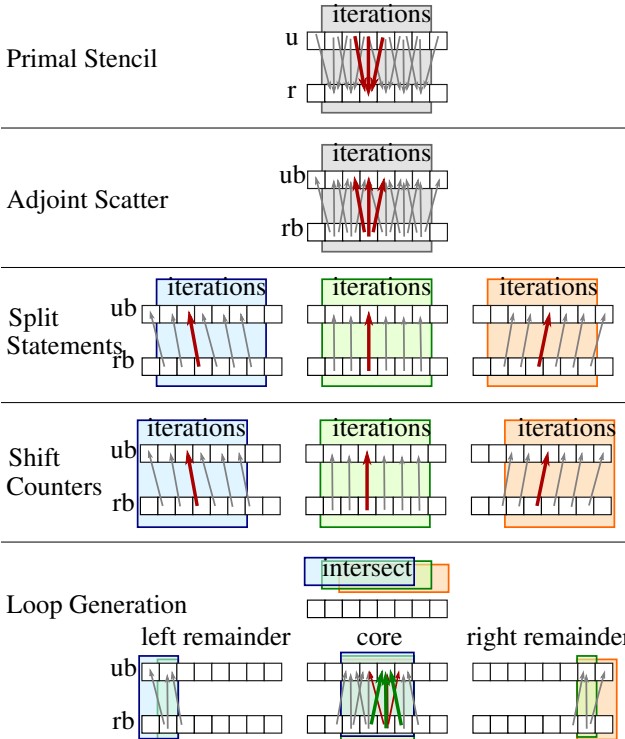

Figure 1: Adjoint stencil transformation step by step, for a one-dimensional three-point stencil. This illustration does not show the read accesses to the primal array u that may still be needed in the derivative code. Red arrows in the primal and adjoint show a group of updates that is performed in the same loop iteration. Those same updates are then highlighted in the split, shifted, and core loop. After regeneration, the red updates are distributed among several iterations, and one iteration in the core loop performs a set of updates as shown by green arrows.

```
  ub[i+1]  += 4.0* c[i] * rb[i];
}
```

We can now substitute the loop counter `i` in the first, second, and third loop with $j:=i-1$, $j:=i$, and $j:=i+1$, respectively, and obtain three loops that each use the loop counter $j$ as write index and have iteration spaces $j \in [0, n-2]$, $j \in [1, n-1]$ and $j \in [2, n]$.

```
for ( j=0; j<=n-2; j++ ) {
  ub[j]  += 2.0 * c[j+1] * rb[j+1];
}
for ( j=1; j<=n-1; j++ ) {
  ub[j]  -= 3.0 * c[j] * rb[j];
}
for ( j=2; j<=n; j++ ) {
  ub[j]  += 4.0 * c[j-1] * rb[j-1];
}
```

We observe that the iteration space of the three loops intersects for $j \in [2, n-2]$. Only the iteration space of the first loop contains $j = 0$, that of the first and second loops contain $j = 1$, that of the second and third contain $j = n-1$, and the iteration space of the third loop contains $j = n$. We can therefore compute the same result using the following parallel loop and remainder statements.

```
ub[0]  += 2.0 * c[1]  * rb[1];
ub[1]  += 2.0 * c[2]  * rb[2];
ub[1]  -= 3.0 * c[1]  * rb[1];
ub[n-1]  -= 3.0 * c[n-1]  * rb[n-1];
ub[n-1]  += 4.0 * c[n-2]  * rb[n-2];
ub[n]  += 4.0 * c[n-1]  * rb[n-1];
#pragma omp parallel for private(j)
for ( j=2; j<=n-2; j++ ) {
  ub[j]  += 2.0 * c[j+1] * rb[j+1];
  ub[j]  -= 3.0 * c[j] * rb[j];
  ub[j]  += 4.0 * c[j-1] * rb[j-1];
```

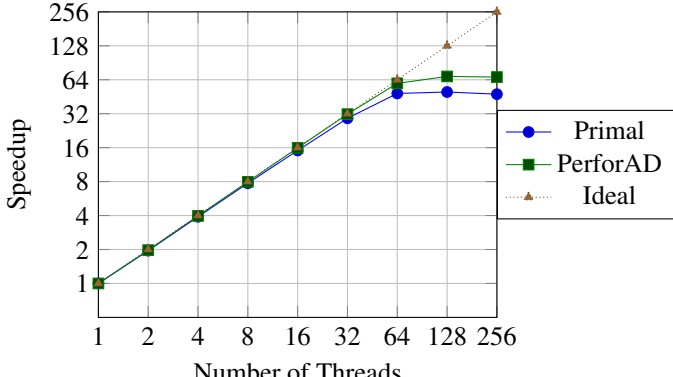

Figure 2: Example of obtained speedups when using PerforAD on a parallelised stencil loop, and executing the result on a KNL processor.

```
}
```

Assuming that $n$ is sufficiently large, the time spent executing the remainder statements will be insignificant compared to that spent inside the loop, which contains only updates to ub[j] that can easily be merged into a single statement to obtain

```
#pragma omp parallel for private(j) shared(rb,ub,c)
for ( j=2; j<=n-2; j++ ) {
  ub[j]  += 4.0 * c[j-1] * rb[j-1]
         - 3.0*c[j] * rb[j]
         + 2.0 * c[j+1] * rb[j+1];
}
```

This adjoint stencil loop has the same set of read and write indices and can be parallelised in the same way as the primal stencil loop. Note that the constant factors $4.0$ and $2.0$ have swapped their position compared with the primal stencil.

## 5   Outlook

In this abstract we motivated the need for an AD tool that can handle performance-optimised programs, and outlined some of the challenges. We then presented PerforAD as an example that successfully navigates this challenge for a very specific kind of input program, namely stencil loops, by applying a predetermined sequence of polyhedral transformations to the derivative program. In our talk we will also show other polyhedral transformations that may be equally beneficial.

While the polyhedral model has advantages, polyhedral compilers are also known to have high compilation times, and the polyhedral model does not naturally capture some things that are essential for AD, such as the dependency relation between reduction variables. Using PerforAD as an example, our talk will illustrate how a polyhedral-like approach that is tailored to AD can overcome these challenges, by creating a derivative program that resembles as closely as possible the structure original program, without exploring the full space of polyhedral transformations.

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
