# OpenReview forum: "Towards Polyhedral Automatic Differentiation"
_NeurIPS.cc/2019/Workshop/Program_Transformations — Program Transformations @NeurIPS2019 Oral_

### Official Review · AnonReviewer1 · 2019-09-26
**Well-written abstract motivating the use of a polyhedral model in AD**

**Confidence:** 4
**Rating:** 8

**Review:**

SIMD parallelism is a cornerstone of ML frameworks and accelerators, and so is AD. This abstract provides a strong motivation for why we should move towards support of SIMD instructions in AD frameworks rather than differentiating higher-level operators. The polyhedral model seems appropriate to use here and is relevant in the context of ML field's recent interest in polyhedral compilers (e.g., Google's MLIR and Facebook's Tensor Comprehensions). I would be eager to hear more about this project at the workshop.

---

### Official Review · AnonReviewer2 · 2019-09-26
**A serious work on how SIMD properties could be pushed from a primal code to its AD-generated gradient code.**

**Confidence:** 4
**Rating:** 8

**Review:**

SIMD qualities of a given code may be lost in its AD-generated gradient code, because reads "become" writes, thus introducing new dependencies/conflicts. The abstract explores a promising approach to overcome this by using loop restructuring/reordering. The abstract generalizes this approach by presenting it in terms of a polyhedral model.
A good abstract presenting a serious work, and right in the scope of the workshop.
My feeling is that this issue is central to many ML systems, as SIMD efficiency is certainly a key to the performance required.
I would encourage the authors to develop the polyhedral aspect further,  especially in the context of Machine Learning where the stencils used are on regular grids (images...) more often than in Scientific Computing (often unstructured meshes).

---

### Decision · Program_Chairs · 2019-10-01

**Decision:**

Accept (Oral)

**Comment:**

This work is relevant to the workshop, spanning the fields of AD and systems/compilers. We are eager to hear more details about the work, although we would like to ask the authors to focus on novel contributions given that similar work on parallelizing reverse mode AD was presented at a previous version of this workshop in 2017.